# Human-like dissociations between confidence and accuracy in convolutional neural networks

**Medha Shekhar** [ORCID]*, **Dobromir Rahnev**

School of Psychology, Georgia Institute of Technology, Atlanta, Georgia, United States of America

* medha@gatech.edu

**Data Availability Statement:** Codes for training the models and performing all the reported analyses are publicly available at https://osf.io/e5d96/.

## Abstract

Prior research has shown that manipulating stimulus energy by changing both stimulus contrast and variability results in confidence-accuracy dissociations in humans. Specifically, even when performance is matched, higher stimulus energy leads to higher confidence. The most common explanation for this effect, derived from cognitive modeling, is the positive evidence heuristic where confidence neglects evidence that disconfirms the choice. However, an alternative explanation is the signal-and-variance-increase hypothesis, according to which these dissociations arise from changes in the separation and variance of perceptual representations. Because artificial neural networks lack built-in confidence heuristics, they can serve as a test for the necessity of confidence heuristics in explaining confidence-accuracy dissociations. Therefore, we tested whether confidence-accuracy dissociations induced by stimulus energy manipulations emerge naturally in convolutional neural networks (CNNs). We found that, across three different energy manipulations, CNNs produced confidence-accuracy dissociations similar to those found in humans. This effect was present for a range of CNN architectures from shallow 4-layer networks to very deep ones, such as VGG-19 and ResNet-50 pretrained on ImageNet. Further, we traced back the reason for the confidence-accuracy dissociations in all CNNs to the same signal-and-variance increase that has been proposed for humans: higher stimulus energy increased the separation and variance of evidence distributions in the CNNs' output layer leading to higher confidence even for matched accuracy. These findings cast doubt on the necessity of the positive evidence heuristic to explain human confidence and establish CNNs as promising models for testing cognitive theories of human behavior.

## Author summary

Humans have the metacognitive ability to reliably express confidence in their decisions. However, certain manipulations can cause confidence to dissociate from accuracy. Particularly, "energy manipulations", where the strength and variance of a stimulus are jointly increased, cause confidence to increase even if accuracy is matched. Typically, these findings have been interpreted as evidence for confidence relying on specific cognitive

**Funding:** This work was supported by the National Institute of Health (award: R01MH119189 to DR) and the Office of Naval Research (award: N00014-20-1-2622 to DR). The funders had no role in study design, data collection and analysis, decision to publish, or preparation of the manuscript.

**Competing interests:** The authors have declared that no competing interests exist.

mechanisms. Alternatively, however, these findings can also be explained purely via changes in perceptual evidence. In this study, we propose that convolutional neural networks (CNN) can offer a way of testing the necessity of these confidence-specific mechanisms in explaining confidence-accuracy dissociations since CNNs lack built-in cognitive mechanisms. We tested different CNN architectures on three kinds of stimulus energy manipulations and found that, similar to humans, the networks also generate robust confidence-accuracy dissociations in all cases. Further, these dissociations emerged solely from changes in the networks' evidence distributions in the output layer where stimulus energy increases the separability as well as variance of evidence distributions. These findings question whether energy-induced confidence-accuracy dissociations can be taken as support for confidence relying on additional mechanisms and suggest the possibility of common, stimulus-driven mechanisms underlying the behavior of humans and CNNs.

## Introduction

Humans have the metacognitive ability to express confidence in their decisions [1,2]. Although confidence is generally reliable in tracking one's performance [3], several kinds of stimulus manipulations have been found to cause confidence to dissociate from accuracy [4–14].

A particular type of confidence-accuracy dissociation has been induced by stimulus manipulations referred to as "energy manipulations" [15,16]. Energy manipulations consist of two independent stimulus features being simultaneously altered in the same direction. Specifically, stimulus energy can be increased by simultaneously increasing stimulus features that provide evidence for the correct choice (e.g., by enhancing stimulus contrast or motion coherence) and increasing stimulus features that provide evidence for the incorrect choice or simply add to the uncertainty of the stimulus (e.g., by enhancing strength of evidence for the distractor stimulus or the variance in the signal). For instance, Herce Castañón et al. (2019) [13] asked observers to judge the mean orientation across an array of Gabor patches. They manipulated stimulus energy by both increasing the contrast of these patches and increasing variance of the orientations across the array. Similarly, Koizumi et al. (2015) [4] asked observers to pick which of two gratings that were superimposed on each other had the higher contrast. They manipulated stimulus energy by simultaneously increasing the contrast of both the target and the non-target gratings. These and similar types of energy manipulations are known to lead to confidence-accuracy dissociations, such that high stimulus energy leads to higher confidence despite accuracy being matched across energy levels [4–9,13]. For simplicity, in the rest of the paper we refer to stimulus features that increase stimulus discriminability as "contrast" and stimulus features that decrease discriminability as "variability" because these terms describe well most designs we examine in this study.

Explanations of these types of dissociations typically invoke cognitive mechanisms or heuristics that are instantiated and tested using modelling frameworks such as signal detection theory (SDT; [17]). The most popular explanation is the positive evidence heuristic which assumes that confidence selectively neglects evidence that disconfirms the observer's choice while the choice is based on a balance of evidence between all choice options [5,15,18–22]. The positive evidence heuristic predicts higher confidence for high-energy stimuli because high-energy stimuli lead to more extreme positive (as well as negative) evidence and confidence ignores negative evidence.

Alternatively, these dissociations have been explained by assuming that humans infer decisions from suboptimal internal models which can result in over-confidence. Particularly,

Herce Castañón et al. (2019) [13] suggested that when observers integrate information across multiple sources (such as an array of Gabor patches), their internal computations are affected by cognitive noise arising from the process of evidence integration. However, observers are "blind" to this cognitive noise, resulting in an internal model that overestimates the quality of evidence and, hence, gives rise to over-confidence. Similarly, Zylberberg et al. (2014 [6], have proposed that observers are insensitive to stimulus variance which also results in an internal model that fails to fully adjust confidence to account for increasing levels of uncertainty, resulting in overconfidence.

On the other hand, a simpler, SDT-based explanation for these effects has also been posited. According to this explanation, energy manipulations lead to changes in the distributions of evidence at the perceptual level, leading to higher confidence [7,23–28]. Specifically, an increase in stimulus energy leads to greater separation between evidence distributions a well as higher overall variability in the observed evidence–which we call the signal-and-variance-increase hypothesis. Consequently, a larger proportion of this distribution is shifted towards extreme values, thus increasing overall confidence [16].

Despite their importance for understanding the processes that give rise to confidence, it has been challenging to adjudicate between these different explanations. The reason is that all explanations can fit the data, but there is no direct way of testing the assumptions inherent in each explanation.

Here, we use convolutional neural networks (CNNs) to distinguish between these competing explanations of the confidence-accuracy dissociations induced by stimulus energy manipulations. Standard CNNs give confidence using the same signal as the decision, and thus do not use the positive evidence heuristic (where decision-incongruent evidence is ignored during confidence computations). They also give confidence without building internal models of the task. Therefore, if the positive evidence heuristic [4,5,18,19] or inference from suboptimal internal models [6,13] are indeed necessary for these dissociations, these networks should fail to mimic human behavior. On the other hand, if confidence-accuracy dissociations arise from a signal-and-variance increase based on inherent stimulus or task characteristics [7,23–27], we can expect neural network models to reproduce human behavior. An additional advantage of CNNs is that, unlike humans, we can directly probe the network's activations and understand the mechanisms underlying their behavior.

In this study, we tested whether three CNN architectures (a custom 4-layer CNN, VGG-19, and ResNet-50) produce human-like confidence accuracy dissociations across three types of energy manipulations. We found that all networks, like humans, expressed higher confidence for higher stimulus energy levels despite accuracies being matched. In addition, they reproduced a dissociation between metacognitive sensitivity and stimulus sensitivity, which is another signature of confidence that is popularly regarded as evidence for the positive evidence bias [18]. Further, we show that the confidence increase in the CNNs was due to an increase in separability and variance of evidence distributions, which is essentially the signal-and-variance-increase hypothesis that has been proposed for humans too [7,16,23–27] These results demonstrate that CNNs exhibit human-like dissociations between confidence and accuracy, showing that it is indeed possible for networks to generate these behaviors in the absence of mechanisms such as the positive evidence heuristic and cognitive inference. Importantly, these observations shed light on the possibility of common mechanisms driven by external features of the environment underlying the behavior of biological and artificial neural networks.

## Results

We tested three CNN architectures (a custom 4-layer CNN, VGG-19, and ResNet-50) on three experiments involving two-choice discrimination judgements about the orientation of stimuli

**Fig 1. Energy manipulations.** In all three experiments, the task involved two-choice discrimination between stimulus configurations oriented clockwise and counterclockwise from the vertical. The upper and lower panels show examples of low- and high-energy stimuli respectively for each experiment. In all three examples, the correct choice is "counterclockwise". (A) Task used by Herce Castañón et al. (2019) [13]. The stimulus consisted of an array of eight noisy Gabor patches with the task involving judgements of mean orientation relative to horizontal. Energy manipulations involved jointly changing the contrast of Gabors as well the variability of orientations across the array. (B) Task used by Koizumi et al. (2015) [4]. The stimulus consisted of two superimposed sinusoidal gratings overlaid by a noise mask. The task was to determine the orientation of the grating with the higher contrast (dominant grating). Increases in energy involved jointly increasing the contrast of the dominant and the non-dominant gratings. (C) The stimulus was a single Gabor patch overlaid with noise and the task was to determine its orientation. Energy was manipulated by jointly changing the contrast and noise level in the patch.

(Fig 1). The deep CNNs–VGG-19 and ResNet-50 –were pretrained on the ImageNet dataset and fine-tuned to perform these tasks. Experiments 1 and 2 have previously been shown to generate confidence-accuracy dissociations in humans [4,13], while Experiment 3 involved a novel task paradigm that has previously not been tested on either humans or neural networks. In each experiment, we manipulated the energy of the stimulus by simultaneously varying two independent stimulus features in the same direction: the contrast of the stimulus and its variability (in Experiments 1 and 3) or the contrast of the stimuli associated with the correct and incorrect choices (in Experiment 2). As is standard in the literature [29], confidence was computed as the output of the final layer transformed by a sigmoid activation function. For all experiments, we trained 25 instances of each network architecture on 10,000 training images over a wide range of stimulus parameters and tested them on 1000 images from each energy condition. The stimulus parameters were chosen such that increasing energy levels resulted in the same average performance level of ~70% across the 25 instances of each network architecture. We targeted a mean accuracy of 70% to avoid floor and ceiling effects on performance while allowing an error rate at which confidence will be informative.

## CNNs exhibit robust confidence-accuracy dissociations

We computed the mean accuracy and confidence of the 25 network instances for each of the three CNN architectures. First, we confirmed that we successfully matched model accuracies across the three energy conditions (Fig 2). Indeed, one-way repeated-measures ANOVAs on model accuracy with energy as factor showed that there were no significant differences in accuracy across the three energy conditions for all experiments and across all three CNN

**Fig 2. Confidence-accuracy dissociations in CNNs.** For all experiments and networks (custom 4-layer CNN, VGG-19 and ResNet-50), accuracy was matched across energy conditions, but confidence significantly increases with energy levels. The violin plots show the kernel density estimates of the data distribution. *p<0.05; **p<0.01; ***p<0.001; ****p<0.0001; n.s., not significant.

architectures: 4-layer CNN (Experiment 1: $F_{(2,24)} = .07$, $p = .93$; Experiment 2: $F_{(2,24)} = 2.80$, $p = .07$; Experiment 3: $F_{(2,24)} = .89$, $p = .42$), VGG-19 (Experiment 1: $F_{(2,24)} = 2.08$, $p = .14$; Experiment 2: $F_{(2,24)} = 1.98$, $p = .15$; Experiment 3: $F_{(2,24)} = .76$, $p = .47$), and ResNet-50 (Experiment 1: $F_{(2,24)} = .60$, $p = .56$; Experiment 2: $F_{(2,24)} = .67$, $p = .52$; Experiment 3: $F_{(2,24)} = .48$, $p = .62$).

On the other hand, we found that increasing stimulus energy led to significantly higher confidence (Fig 2). Indeed, one-way repeated-measures ANOVAs showed significant differences in model confidence across the three energy conditions for all experiments in each of the three network architectures: 4-layer CNN (Experiment 1: $F_{(2,24)} = 36.27$, $p < .0001$; Experiment 2: $F_{(2,24)} = 189.77$, $p < .0001$; Experiment 3: $F_{(2,24)} = 51.82$, $p < .0001$), VGG-19 (Experiment 1: $F_{(2,24)} = 85.60$, $p < .0001$; Experiment 2: $F_{(2,24)} = 155.53$, $p < .0001$; Experiment 3: $F_{(2,24)} = 39.15$ $p < .0001$), and ResNet-50 (Experiment 1: $F_{(2,24)} = 20.75$, $p < .0001$; Experiment 2: $F_{(2,24)} = 175.36$, $p < .0001$; Experiment 3: $F_{(2,24)} = 19.85$, $p < .00001$). Further, pairwise comparisons between the low and high energy levels showed significant increases in confidence for the high energy condition for all networks and all three experiments (all p's $< 0.0001$; S1 Table and Fig 2). These results mimic findings from human behavioral studies where increasing the energy of the stimulus leads to increases in confidence, despite accuracies being matched across conditions [4–14]. These findings cast doubt on the necessity of the positive evidence and noise-blindness hypotheses for explaining confidence.

We note that in most cases, the CNNs exhibit average confidence levels greater than 0.9 despite mean accuracy being around 70%. These findings are in line with observations that neural networks, particularly CNNs, often exhibit overconfidence in their responses [29,30]. Nevertheless, we verified that the network's confidence is informative and can reliably discriminate between correct and incorrect choices (S1A Fig). We also confirmed that these effects remain after calibrating the networks' confidence using temperature-scaling (S1B Fig; [29,30].

## Mechanism behind the confidence-accuracy dissociation in CNNs

The finding of confidence-accuracy dissociations in CNNs suggest that the positive evidence and noise-blindness hypotheses are unnecessary to explain these effects. However, to test whether the alternative signal-and-variance-increase process underlies this behavior, we examined how changes in stimulus energy affect the CNNs' evidence distributions. Specifically, the signal-and-variance-increase hypothesis predicts that increasing stimulus energy leads to greater separation of evidence between the two stimulus categories as well as an increase in the variance of evidence. As a result of these changes, the evidence distributions shift towards more extreme values, leading to higher confidence overall. Therefore, we probed the CNNs' evidence distributions to examine whether changes in the networks' evidence are consistent with this hypothesis.

We aggregated the activations generated in the networks' output layer in response to images for each stimulus category separately for each energy condition. We refer to these activations as "evidence" because they refer to the final activations generated within the network for a given choice option before the classification response is given. We plotted these activations for each stimulus category ($S_1$ for counterclockwise stimuli and $S_2$ for clockwise stimuli) and observed how the characteristics of these distributions vary across energy levels. We quantified the separation between the two stimulus categories as the distance between their means ($\mu_{S_2} - \mu_{S_1}$; where $\mu_{S_i}$ refers to the mean of the evidence distribution for stimulus category $i$) and the spread of distributions as the average standard deviation (SD) of the two evidence distributions. The separation between distributions and the average SD was computed separately for each of the 25 network instances.

We found that increasing energy levels led to larger separation between the $S_1$ and $S_2$ evidence distributions as well as an increase in the variance of these distributions (**Fig 3**). Indeed, one-way repeated measures ANOVAs showed significant differences in the separation between the two distributions between the three energy conditions for all experiments and networks: 4-layer CNN (Experiment 1: $F(2,24) = 320.83$, $p < .0001$; Experiment 2: $F(2,24) = 342.21$, $p < .0001$; Experiment 3: $F(2,24) = 92.32$, $p < .0001$), VGG-19 (Experiment 1: $F(2,24) = 169.78$ $p < .0001$; Experiment 2: $F(2,24) = 177.81$, $p < .0001$; Experiment 3: $F(2,24) = 160.13$, $p < .0001$), and ResNet-50 (Experiment 1: $F(2,24) = 444.34$, $p < .0001$; Experiment 2: $F(2,24) = 121.60$, $p < .0001$; Experiment 3: $F(2,24) = 54.37$, $p < .00001$). Further, pairwise comparisons using two-sided t-tests showed a significant increase in the separation between the distributions for the two stimulus categories from the low- to high-energy conditions for all experiments and networks (all p's $< 0.0001$; **S1 Table**).

Parallel to the results on separability, we found that higher stimulus energy also led to increases in the variance of the evidence distributions (**Fig 3**). Indeed, one-way ANOVAs on the average SD of evidence distributions also yielded significant differences between the three energy conditions for all experiments and networks: 4-layer CNN (Experiment 1: $F(2,24) = 408.38$, $p < .0001$; Experiment 2: $F(2,24) = 284.13$, $p < .0001$; Experiment 3: $F(2,24) = 120.15$, $p < .0001$), VGG-19 (Experiment 1: $F(2,24) = 140.25$, $p < .0001$; Experiment 2: $F(2,24) = 108.84$, $p < .0001$; Experiment 3: $F(2,24) = 177.82$ $p < .0001$), and ResNet-50 (Experiment 1: $F(2,24) = 229.60$, $p < .0001$; Experiment 2: $F(2,24) = 108.84.38$, $p < .0001$; Experiment 3: $F(2,24) = 69.99$, $p < .00001$). Further, pairwise t-tests across low and high energy conditions revealed significant increases in the average SD of activations for all networks and across all three experiments (all p's $< 0.0001$; **S1 Table**).

These findings show that the changes in the CNN's evidence distributions are indeed consistent with the signal-and-variance-increase hypothesis. How do these shifts in evidence distributions lead to higher confidence for higher energy levels? The concurrent increase in separation between the evidence for the two stimulus categories and the variability of evidence ensures that the networks' overall ability to discriminate between the two stimulus classes remains constant between the three energy conditions. Specifically, any improvement in the networks' performance yielded by the increased separation between the evidence distributions for the two categories is counteracted by the evidence itself becoming more variable and confusable between the two classes. Nevertheless, confidence differences still emerge between conditions because the higher variance and separation between these distributions results in larger proportions of evidence being pushed towards extreme values that get assigned higher confidence. Together, these findings show that the confidence-accuracy dissociations produced by the CNNs can indeed be explained by the signal-and-variance-increase hypothesis.

## Different stimulus features selectively influence the separability and spread of evidence distributions

While changes in the evidence distributions can explain the differences in confidence between energy conditions, it is still unclear why energy manipulations have this effect on the evidence distributions. Energy manipulations consist of two independent stimulus variables being manipulated simultaneously in the same direction such as its contrast and variability. To understand each variable's individual effect on the networks' evidence distributions, we manipulated each variable separately (while keeping the other fixed). We then compared these changes to the changes in evidence distributions observed during energy manipulations. For Experiments 1 and 3, the two variables manipulated were the contrast and variability of the stimuli, whereas in Experiment 2, the variables manipulated were the contrast of the correct vs.

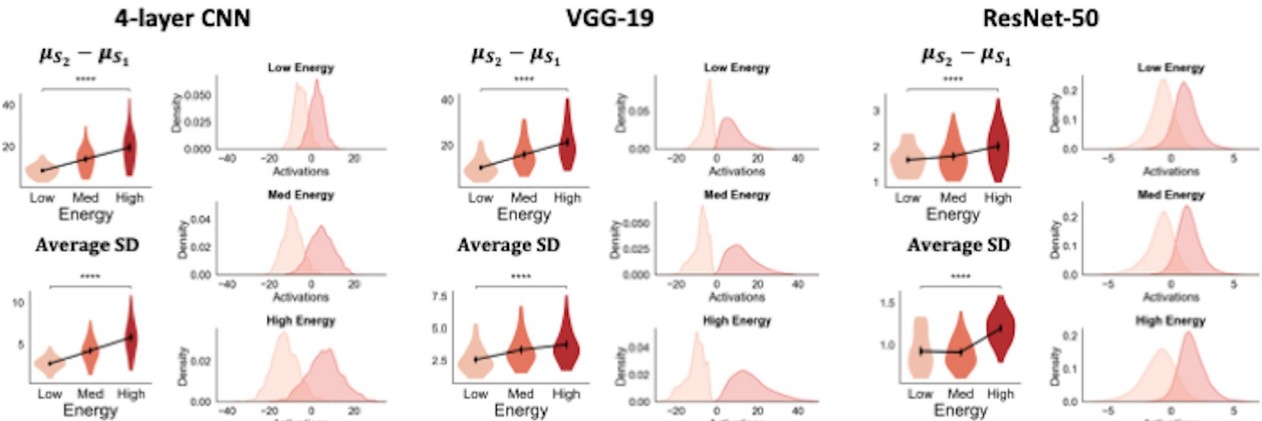

**Fig 3. Energy manipulations increase both the separability and variance of the networks' output layer activations.** For all three experiments, the separability between the distributions of evidence for the two stimulus categories, as well as the variance of the evidence distributions, increased with energy levels. For each network, the figure shows the distance between the $S_1$ and $S_2$ evidence distributions and their standard deviations (SD) across the 25 model instances. We note that these networks appear to represent evidence on their own unique internal axis. Therefore, to optimally visualize differences in their evidence distributions, each network's distributions have been plotted on their own unique scale. The kernel density plots show the distribution of activations aggregated across all 25 network instances. *p<0.05; **p<0.01; ***p<0.001; ****p<0.0001; n.s., not significant.

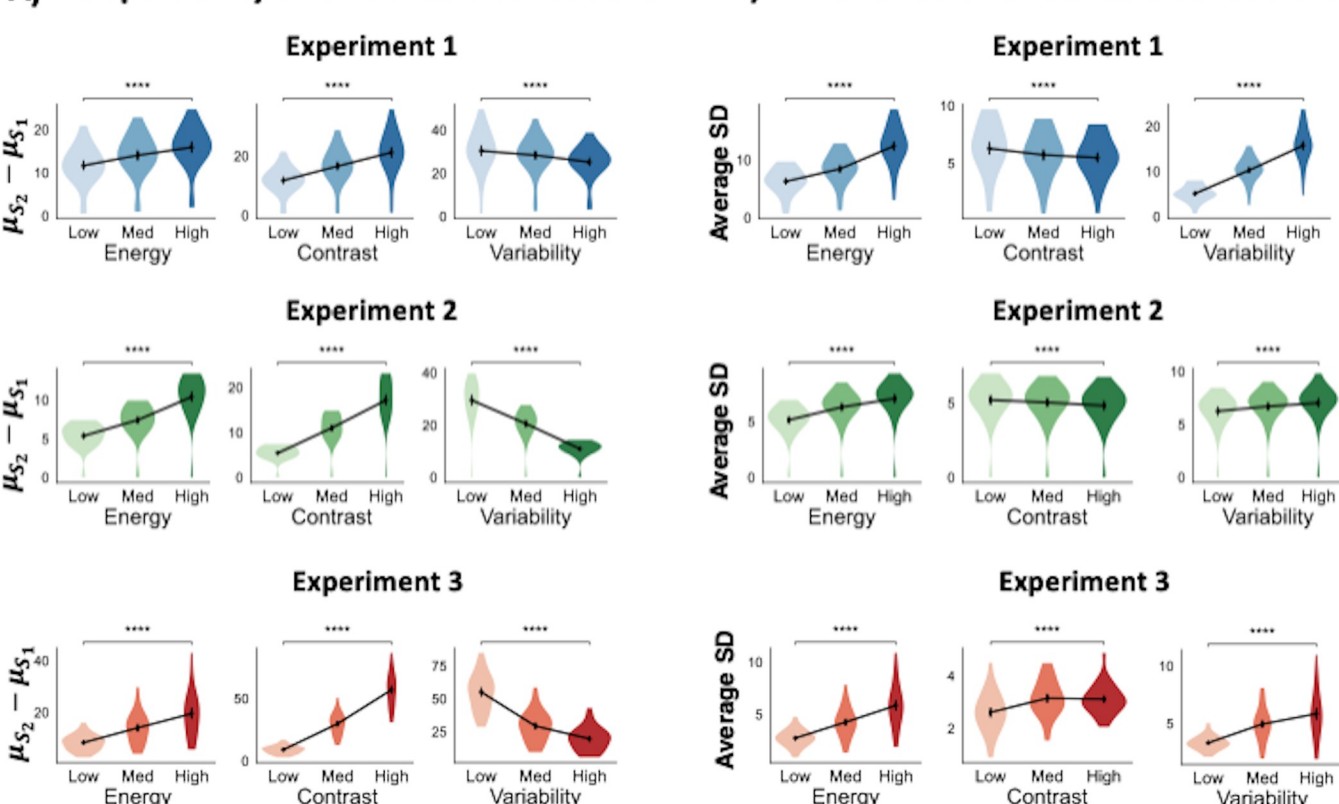

**Fig 4. Changes in the separation and spread of evidence distributions induced by energy, contrast, and variability manipulations.** The plots show A) the average distance between the mean activations for $S_1$ and $S_2$ stimuli (left) and B) the average standard deviation (SD) of activations (right) in the final layer of the shallow CNNs for Experiments 1–3 in response to changes in stimulus energy, contrast and variability. Note that the energy results in both panels are equivalent to the 4-layer CNN results from Fig 3. For all experiments, increasing energy and contrast levels increases the separation between the two stimulus categories, while increasing variability decreases the separability between the two stimulus categories. On the other hand, increasing stimulus energy and variability increases the spread of evidence distributions, while increasing contrast decreases the spread of evidence for all experiments except Experiment 3 (where increasing contrast increases the spread of evidence). These results suggest contrast and noise changes selectively drive changes in separation and variance of evidence distributions respectively. The violin plots show the kernel density estimates of the data distribution. *p<0.05; **p<0.01; ***p<0.001; ****p<0.0001; n.s., not significant.

incorrect grating. For conciseness, we refer to manipulations of both variability and the contrast of the incorrect grating as manipulations of variability. For each experiment, we investigated the networks' activations separately for manipulations of contrast and manipulations of variability.

We first examined the effects of contrast and variability manipulations on the separability of the evidence distributions. We found that while increasing stimulus contrast increased the separation between stimulus categories, increasing variability led to a decrease in their separation (**Fig 4A**). Indeed, there were significant mean differences in evidence separability for both manipulations of contrast (Experiment 1: $F(2,24) = 233.44$, $p < .0001$; Experiment 2: $F(2,24) = 283.89$, $p < .0001$; Experiment 3: $F(2,24) = 469.16$, $p < .0001$) and manipulations of variability (Experiment 1: $F(2,24) = 57.75$, $p < .0001$; Experiment 2: $F(2,24) = 324.00$, $p < .0001$; Experiment 3: $F(2,24) = 763.58$, $p < .0001$). Pairwise comparisons showed that contrast significantly increased separability between stimulus categories (all p's $< .0001$, paired t-tests comparing the lowest and highest contrast levels), while variability significantly decreased this separation (all p's $< .0001$, paired t-tests comparing the highest and lowest contrast levels; **S2 Table**).

These results suggest that the increase in separability between evidence distributions observed during energy manipulations is primarily driven by changes in stimulus contrast.

We then examined the effects of contrast and variability manipulations on the spread of the evidence distributions. We found that increasing stimulus contrast decreased the variance of evidence distributions, while increasing variability increased their variance. Indeed, there were significant differences in the spread of evidence distribution for both manipulations of contrast (Experiment 1: $F(2,24) = 56.03$, $p < .0001$; Experiment 2: $F(2,24) = 21.71$, $p < .0001$; Experiment 3: $F(2,24) = 32.47$, $p < .0001$) and manipulations of variability (Experiment 1: $F(2,24) = 388.45$, $p < .0001$; Experiment 2: $F(2,24) = 48.16$, $p < .0001$; Experiment 3: $F(2,24) = 53.28$, $p < .0001$). Pairwise comparisons showed that variability significantly increased the spread of distributions (paired t-tests comparing the highest and lowest contrast levels; all p-values $< .0001$; S2 Table) while contrast significantly decreased their spread for Experiments 1 and 2 (paired t-tests comparing the highest and lowest contrast levels; all p-values $< .0001$; S2 Table). For Experiment 3, however, increasing contrast significantly increased the spread of evidence distributions ($p < 0.001$). These results suggest that the increase in variability of evidence observed during energy manipulations is primarily driven by changes in stimulus variability (Fig 4; right). Overall, these results demonstrate that manipulations of stimulus contrast and variability have opposite effects on the separability and spread of activations in CNNs' final layer, such that contrast manipulations have a larger effect on separability and variability manipulations have a larger effect on spread. Thus, combining both manipulations in a single "energy" manipulation in these networks leads to both increased separability and spread of the evidence distributions.

## CNNs can reproduce dissociations between type-1 and type-2 sensitivity typically regarded as evidence for the positive evidence mechanism

So far, our results have shown that CNNs, in spite of lacking the positive evidence mechanism, can produce the kind of stimulus-energy induced confidence-accuracy dissociations that have typically been attributed to the positive evidence heuristic in humans. Another feature of confidence that has been attributed to the positive evidence mechanism is the observation that under certain conditions, an observer's stimulus sensitivity (ability to discriminate between stimulus classes) and metacognitive sensitivity (ability to discriminate between correct and incorrect responses using confidence) are found to dissociate from each other under certain experimental designs [18]. Using SDT, one can quantify stimulus sensitivity as the amount of information available for the primary stimulus judgement (d'), and metacognitive sensitivity as the amount of information underlying confidence judgements (meta-d'). Typically, an increase in d' translates into a proportional increase in meta-d' [31]. However, Maniscalco et al. (2016) [18] found that under a certain task paradigm, these two measures can dissociate from each other. More specifically, the paradigm consists of a two-choice discrimination task, where the contrast of one stimulus category ($S_1$) is held constant while the contrast of the other stimulus is allowed to vary over discrete levels ($S_2$). Under these conditions, meta-d' decreases with d' for trials where the observer responds "$S_1$," but meta-d' increases with d' for trials where the observer responds "$S_2$." Importantly, this effect was explained by a model incorporating the positive evidence mechanism, whereas a competing model that assumed equal weights for positive and negative evidence failed to account for this behavior [18]. These findings are typically regarded as evidence for the existence of the positive evidence mechanism for confidence.

To further test the necessity of the positive evidence mechanism in explaining confidence, we assessed whether CNNs (which lack this mechanism) can also reproduce the dissociation

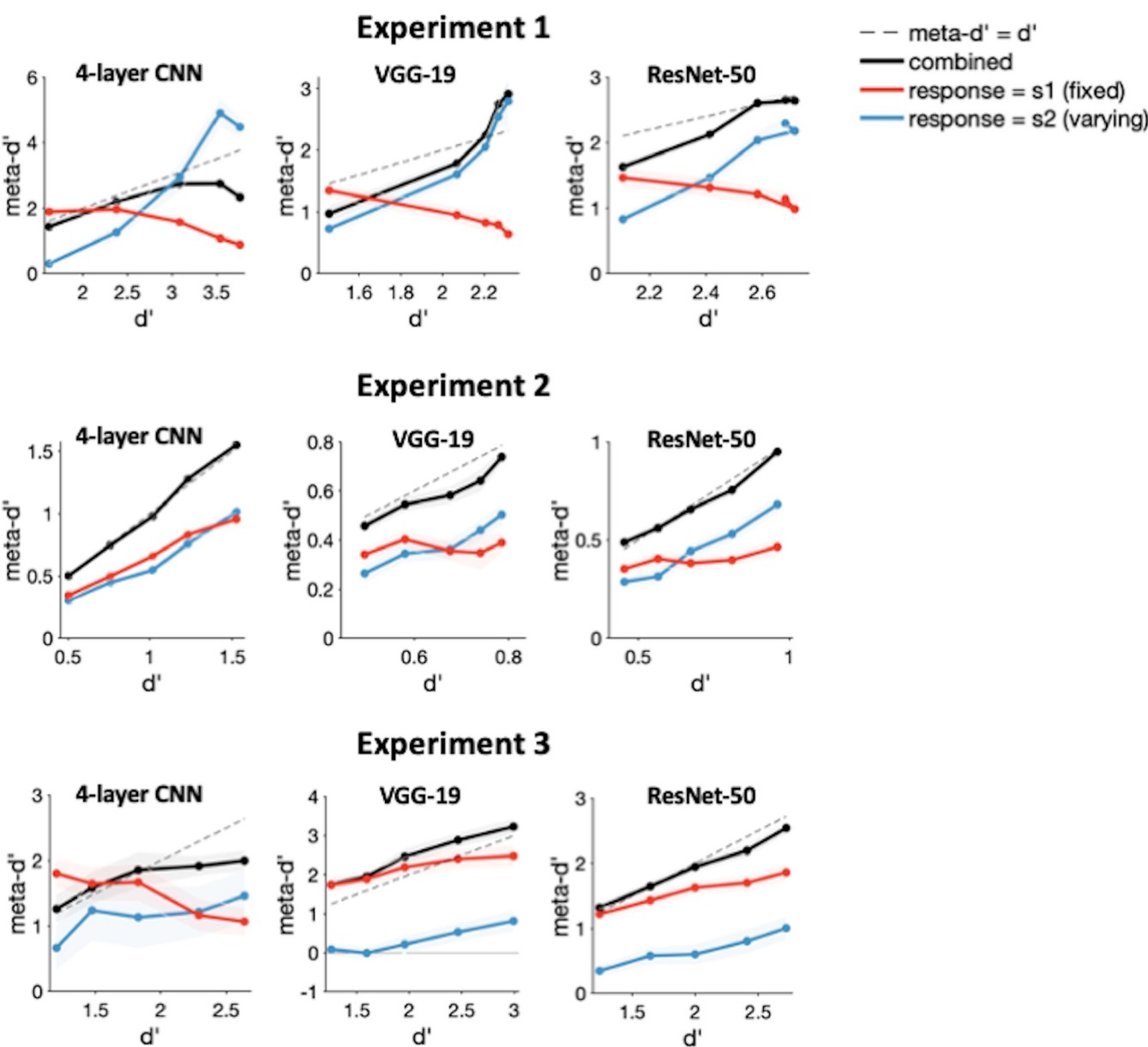

**Fig 5. Dissociations between meta-d' and d'.** We tested the three networks (4-layer CNN, VGG-19 and ResNet-50) on the task paradigm from Maniscalco et al. (2016) [18] which demonstrated a dissociation between d' and meta-d' when the contrast of one stimulus category ($S_1$) remains fixed while the contrast of the other stimulus is increased in discrete steps ($S_2$) For this design, meta-d' increases with d' as expected for trials in which the observer responds "$S_2$", but meta-d' decreases with d' for trials where the observer responds "$S_1$". Maniscalco et al. (2016) [18] showed that this behavioral effect can be explained by a model incorporating the positive evidence bias. Here, we simulated this task paradigm for the stimuli in Experiments 1–3. The responses generated by our networks show that they can indeed generate the meta-d'/d' dissociations observed in humans for at least two out of three experiments. While the 4-layer network fails to reproduce this behavior for Experiment 2, VGG-19 and ResNet-50 fail to produce this behavior for Experiment 3, suggesting that these dissociations may depend on specific interactions between the stimuli and the networks.

observed in Maniscalco et al. (2016) [18]. We simulated the above task paradigm for our previously trained networks across the three experiments (as done by Webb et al., 2023 [28] and found that all the networks were indeed able to reproduce a clear dissociation between meta-d' and d' for at least two out of the three experiments (**Fig 5**). While the 4-layer CNN produced this dissociation for Experiments 1 and 3, VGG-19 and ResNet-50 produced these

dissociations for Experiments 1 and 2. Specifically, meta-d' increases with d' for trials with "$S_2$" responses (where the contrast of $S_2$ varies across trials), but meta-d' decreases with d' for trials with "$S_1$" responses (the contrast of "$S_1$ remaining fixed across trials), producing the distinct cross-over signature shown by Maniscalco et al. (2016) [18] It is possible that unlike the confidence-accuracy dissociations examined above, meta-d'/d' dissociations may partially rely on a different set of processes that are more sensitive to the specifics of the stimuli and network architectures. However, it is not yet clear what these processes may be and why meta-d'/d' dissociations are less robust than confidence-accuracy dissociations. Further work is required to address these questions. Nevertheless, our findings demonstrate that for at least for some stimulus manipulations and networks, the positive evidence mechanism is unnecessary to explain effects that have typically been considered as evidence for this mechanism.

## Confidence accuracy dissociations in CNNs generalize across stimulus paradigms but do not always mimic human behavior

Our results demonstrate that CNNs can produce human-like confidence-accuracy dissociations where confidence increases with increasing stimulus energy levels. However, when using color stimuli, energy manipulations have been found to *decrease* confidence while accuracy remains matched across conditions [8–12]. These findings have been explained by assuming a mechanism of "robust averaging" where highly atypical stimuli are down-weighted in the final decision [32,33]. Our findings thus far establish that simple changes in the separation and variance of stimulus evidence distributions can explain human behavior that has typically been attributed to cognitive mechanisms such as the positive evidence bias and noise-blindness. Here, we sought to further test whether the "robust averaging" mechanism can also be realized through such changes in evidence distributions.

Following the same procedure as done previously, we tested our CNNs on the task from Boldt et al. (2017) [11] where subjects identified whether the mean color across an array of eight colored patches was closer to red or blue (**Fig 6A**). Energy manipulations involved jointly increasing the intensity of the color ("blueness" or "redness" of stimuli) and the variance of color across the eight patches. The stimulus parameters were chosen such that we obtained matched average accuracy levels of ~70% across the three energy levels. A one-way repeated measures ANOVA showed no significant mean differences in accuracy between the three energy conditions for all three networks– 4-layer CNN (F(2,24) = .78, p = .47), VGG-19 (F (2,24) = .31, p = .74) and ResNet-50 (F(2,24) = .15, p = .86).

However, the ANOVA revealed significant differences in mean confidence between the three stimulus energy levels for all three networks (4-layer CNN: F(2,24) = 152.50, p < .0001; VGG-19: F(2,24) = 57.72, p < .0001; and ResNet-50: F(2,24) = 68.71, p < .0001; pairwise comparisons using t-tests between low and high energy levels: all p's < 0.0001; **Fig 6B** and **S1 Table**). Further, these behavioral effects were associated with increases in both the separability and variance of evidence distributions (all pairwise t-tests between low and high energy levels < .002; **Fig 6C** and **S2 Table**), in line with the signal-and-variance-increase effect. While these results are consistent with findings from Experiments 1–3, they fail to replicate human behavior suggesting that the signal-and-variance-increase process cannot account for all types of confidence-accuracy dissociations, particularly, the ones attributed to a process of "robust averaging". Future studies must investigate whether incorporating the robust averaging mechanism can restore human-like confidence behavior in CNNs. Nevertheless, these results establish the generalizability of the signal-and-variance-increase process underlying confidence-accuracy dissociations in CNNs. Further, they suggest that while for some stimuli (such as the Gabor patches tested here), the signal-and-variance-increase hypothesis can possibly explain

## A)

### Color discrimination experiment

**Low Energy stimuli**
Low color intensity + Low color variance

**High Energy stimuli**
High color intensity + High color variance

**Mean color:**
**Blue**

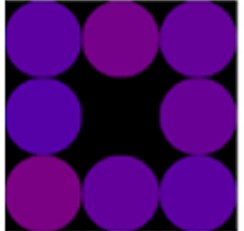

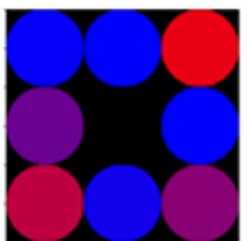

## B)

### Accuracy and confidence

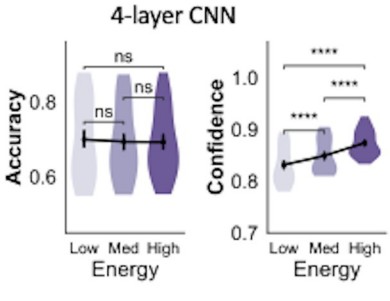

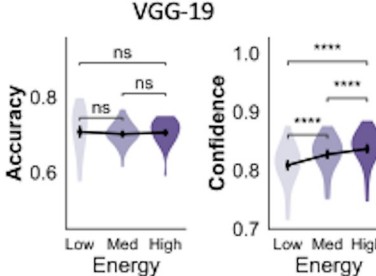

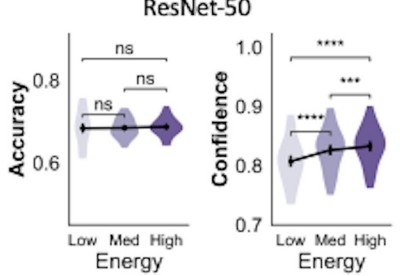

## C)

### Output-layer activations of the network

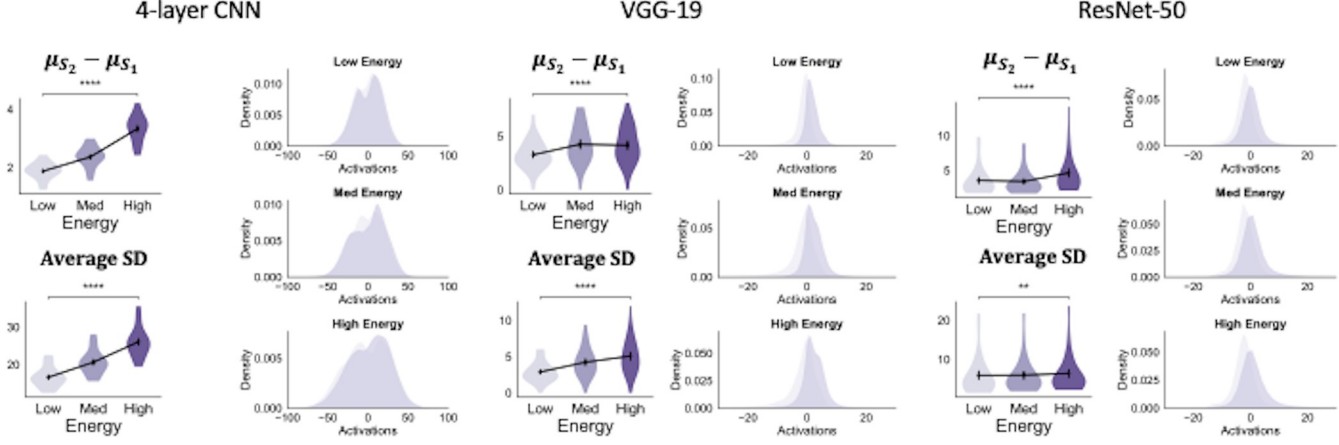

**Fig 6. Confidence-accuracy dissociations in a color discrimination task.** (A) The stimulus consisted of an array of eight colored circles. The task was to determine whether the mean color across the eight patches was more blue or red. In this example, the mean color is more blue than red. Energy manipulations involved joint changes to the intensity of color (the amount of "blueness" or "redness" of the patches as well the variance in color across the array. (B) The networks' accuracy was matched across energy conditions, but confidence significantly increased with energy levels. (C) The separability between the stimulus categories as well as the variance of the evidence distributions increased with energy levels for all three networks. The panels on the top-left for each network show the average distance between the $S_1$ and $S_2$ evidence distributions across the 25 model instances. The panels on the bottom-left show the average standard deviation (SD) across the two distributions across all model instances. The panels on the right show the distribution of activations aggregated across all 25 network instances. *$p<0.05$; **$p<0.01$; ***$p<0.001$, ****$p<0.0001$; n.s., not significant.

the behavior of both humans and CNNs, for certain other stimuli (such as colors), the behavior of CNNs and humans diverge likely due to human visual decision-making relying on additional, stimulus-specific mechanisms that CNNs lack.

## Discussion

We found that convolutional neural networks (CNNs) robustly produce human-like confidence-accuracy dissociations in response to stimulus energy manipulations. In humans, these dissociations have been taken as evidence for the existence of specific mechanisms such as the positive evidence heuristic [4,5,20] and noise blindness [6,13]. Since CNNs lack such built-in mechanisms, their ability to mimic human confidence behavior implies that these popular theories are unnecessary to explain energy-induced confidence-accuracy dissociations. Instead, our findings support the idea that changes in the evidence distributions are sufficient to account for such dissociations. Indeed, we find that in CNNs, these dissociations are explained by the fact that higher stimulus energy makes their evidence distributions both more separable and more variable. These findings suggest the possibility of common, stimulus-based processes driving the behavior of both artificial and biological systems and demonstrate the usefulness of CNNs in testing the necessity of specific computational explanations of human behavior.

### Implications for the positive evidence bias in confidence

The positive evidence (PE) heuristic is one of the most popular proposals regarding the computations underlying confidence [4,5,15,18–20]. Despite its popularity, however, findings from recent studies suggest that the positive evidence bias may not be necessary to explain confidence [28,34,35].

First, the previous studies that found support for the PE bias did not perform extensive model comparisons. Rather, the PE model was typically compared to a model which assumed that confidence is based on a balance of evidence between the two choice options [18,19]. Importantly, in this model, stimulus manipulations were assumed to have no effect on the variance of the internal distributions of evidence. Further, the PE model has rarely been compared to other recently developed models of confidence in the literature [34,36–43]. When such model comparisons have been performed, the PE model usually ranks poorly relative to models that allow suboptimalities in confidence to manifest via other mechanisms such as metacognitive noise or the visibility heuristic [34,35].

Second, behavioral evidence for the PE bias mainly rests on two observed patterns in confidence–increase in confidence with stimulus energy despite matched accuracies [4,5,20] and the dissociation between d' and meta-d' under a certain stimulus paradigm [18]. However, these studies have not considered the existence of alternative mechanisms that can explain these behavioral effects. Here, we show that CNNs, which lack any positive evidence-specific mechanisms, can produce both these signatures via changes in their evidence representations. This result thus questions whether such behavioral findings can, by themselves, be taken as evidence for the positive evidence mechanism.

Finally, the conclusion that the positive evidence bias is not necessary to explain confidence is supported by a recent paper that also tested how CNNs behave in the context of energy manipulations [28]. These CNNs were specifically trained to compute confidence optimally, but nonetheless reproduced both behavioral signatures of the PE bias–confidence-accuracy dissociations and the meta-d'/d' dissociations. Further analyses suggested that these dissociations were being driven by the statistical structure of the training data. Critically, a Bayesian model whose internal evidence distributions reflected the true stimulus structure could not

only reproduce all the observed dissociations but also predict the network's responses better than the positive evidence model. Finally, in line with our results, they also found that energy manipulations increased the separation and variance of the latent stimulus distributions and confirmed that even networks not explicitly trained to give confidence (such as the ones we test here) can produce these dissociations. In sum, Webb et al.'s [28] findings challenge the PE mechanism and also reveal how the statistics of the stimuli alone might influence a system's learned representations to drive confidence dissociations.

## Implications for other theories of confidence

Findings of energy-induced confidence-accuracy dissociations have also been interpreted as evidence for other processes underlying confidence judgments. For instance, Herce Castañón et al. (2019) [13] argued that a range of suboptimal behaviors arise from noise blindness, such that observers neglect to account for the noise arising from their own cognitive computations when integrating across variable evidence samples. This noise blindness results in observers failing to adjust their responses to increasing levels of uncertainty. In addition to being over-confident for high-energy stimuli, Herce Castañón et al. [13] reported that observers neglect stimulus base rates in the high-energy condition and thus fail to appropriately shift their decision criterion in favor of the more frequent stimulus. However, the signal-and-variance-increase hypothesis is sufficient to explain both the suboptimal behaviors they report. According to the signal-and-variance-increase hypothesis, when the separation between the distributions and their variance is high, a criterion shift of same magnitude will have a smaller effect on choice probabilities compared to when the distributions have low separation and variance, thus appearing as if the observers have failed to shift their criteria appropriately. Indeed, simulations of criterion shifts under the signal-and-variance-increase hypothesis reproduced their reported effects (S2 Fig).

However, it must be noted that overconfidence and base-rate neglect both arise from the observers' failure to scale their decision and confidence criteria in response to changing levels of uncertainty in stimulus evidence. In that sense, this effect might reflect a blindness to changes in the properties of evidence at the decision stage. Nevertheless, this mechanism is distinct from the one proposed by Herce Castañón et al. (2019) [13] because in their model the blindness is towards noise arising from the observers' own internal cognitive processes, rather than towards noise arising from stimulus-driven changes in sensory evidence.

## Evidence for the signal-and-variance-increase hypothesis

The notion that confidence can be influenced solely by changes in sensory evidence is not new. Brain stimulation applied to lower [25,26] and mid-level visual areas [27] has been found to affect confidence, independent of changes in accuracy. These effects were captured well by SDT models where stimulation increased the variance of internal evidence. Other task manipulations involving attention [23,24] and evidence volatility [7] also produced similar dissociations in confidence that were accounted by SDT modelling via changes in the trial-by-trial variance of sensory evidence. Our current findings corroborate these findings and extend them by adding energy manipulations to the list of factors that can produce confidence-accuracy dissociations via the signal-and-variance-increase process. However, it must be noted that this process by itself does not provide a mechanistic explanation for why changes in stimulus energy should lead to changes in the properties of the evidence distributions. Further research is required to characterize the mechanisms responsible for the observed relationships between stimulus features and the perceptual evidence they generate.

## CNNs as models for understanding human vision

Several recent studies have argued that deep neural networks can provide meaningful insights into the goals and constraints that have shaped human perception [44–52]. Our findings carry implications about the external constraints that may have shaped visual processing. For instance, we find that both humans and CNN models produce the same behavior in response to energy manipulations and, further, the same signal-and-variance-increase process posited to work in humans underlies the behavior of CNNs. One possible explanation for the similarity in feature processing between the two networks is that the processing is being driven by the stimulus feature itself, rather than by some mechanism inherent to the network.

Our study can also help inform the debate regarding when CNNs can serve as useful models of human perceptual decision making. Bowers et al. (2023) [53] argue against the notion that CNNs can be applied as models of human perception since these models have not been adequately tested on their ability to account for findings in cognitive science research. Although CNNs perform visual classification tasks with high levels of accuracy, it is not yet understood whether these networks process stimulus features in ways similar to humans. Bowers et al. [53] argue that to establish such similarity in processing, it is important to test CNN's behavior on manipulations of independent stimulus features and compare their behavior to humans on psychological tasks. In this study, we follow a similar logic as we manipulate stimulus features such as contrast and variance and find that CNNs indeed respond similarly to humans under the specific stimuli tested here. Further, these manipulations change the properties of the CNNs' evidence distributions in a way that is consistent with the predictions of an empirically validated model of perception. We suspect that, in the case of the stimuli tested here, a common external factor, such as the statistical properties of the stimuli themselves, drive the behavior of both humans and ANNs. If so, this suggest that in tasks where performance is strongly driven by the structure of the stimuli themselves, CNNs may provide a useful model of human perceptual decision making.

## Other types of confidence-accuracy dissociations

In the current study, we tested CNNs on a specific type of confidence-accuracy dissociation induced by energy manipulations. However, prior research has found that an abundance of factors can cause confidence to dissociate from accuracy. Some of these include motor preparation and execution [54–56], transcranial magnetic stimulation [25,57–60], differences in pre-stimulus brain activity [61,62] confidence history [63,64], attention [24,65], arousal [66] and stimulus visibility [34,67]. However, in this study, we only chose to test energy-induced confidence accuracy dissociations as these manipulations can be readily applied to CNNs unlike those involving motor preparation, transcranial magnetic stimulation, attention, arousal, etc. Future studies can test the proposed cognitive mechanisms underlying other kinds of confidence-accuracy dissociations against suitable alternative explanations involving only changes to perceptual evidence to gain insight into the true mechanisms of confidence. However, in the case that CNNs fail to mimic human behavior, one cannot automatically infer that the mechanism being tested is necessary to produce that behavior. Such divergence in the behavior of humans and CNNs can alternatively be explained by fundamental differences in how the two systems process visual stimuli themselves (Wichmann & Geirhos, 2023).

Importantly, beyond the examples of high-energy stimuli leading to high confidence examined in this paper, there are two kinds of stimuli that break that rule. For these stimuli, energy manipulations lead to confidence that decreases with energy levels. Firstly, Spence et al. (2016, 2018) [9,10] observed this effect for random dot motion stimuli. It is possible to explain this effect by assuming that increasing the variance of motion direction may deliver high-level cues

regarding task difficulty. In turn, subjects may use these difficulty cues to decrease their confidence. Since our CNNs do not work on dynamic, dot motion stimuli, we could not test them on these stimuli. Secondly, Boldt et al. (2017, 2019) and Desender et al. (2018) [11,12,14] found a similar effect for arrays of colored dots. When we tested our CNNs on these color stimuli (**Fig 6**), we found that changes in evidence distributions alone cannot account for these effects, and thus it is likely that these manipulations engage other cognitive mechanisms. Indeed, these color tasks have been proposed to trigger "robust averaging" where observers down-weight highly atypical evidence samples (32). Since high-energy stimuli generate more extreme evidence, ignoring (or down-weighting) them leads to a lower overall estimate of evidence for confidence. Future studies should test whether incorporating high-level cues about task difficulty and the robust averaging mechanism into CNNs can indeed generate this effect.

## Conclusion

In this study, we demonstrate that CNNs can generate human-like confidence-accuracy dissociations in response to stimulus energy manipulations via changes in the variance and separability of the evidence distributions in their output layer. These findings cast doubt on the necessity of invoking several specific explanations for this phenomenon–particularly the popular assumption that confidence is derived from a positive evidence heuristic. Our results highlight the necessity of disentangling perceptual and cognitive explanations of behavior and establish CNNs as promising models for testing the necessity of cognitive explanations of human behavior.

## Methods

### Stimuli and task

We tested several convolutional neural networks on three main experiments. The task paradigms for Experiments 1 and 2 were adapted from Herce Castañón et al. (2019) [13] and Koizumi et al. [4]. Both of these papers found confidence-accuracy dissociations in humans where confidence was found to increase with stimulus energy levels. However, we modified their original tasks with the purpose of isolating the effects of stimulus energy. Specifically, the task in Herce Castañón et al. (2019) [13] was designed in a 2x2 factorial manner with stimulus contrast and tilt variability as factors and participants were presented with cues at the beginning of each trial that indicated which stimulus category was more likely to be presented. The task in Koizumi et al. (2015) [4] was also a 2x2 factorial design consisting of manipulations of both positive evidence and task difficulty. In our simulations, we simplified these designs by excluding the cue manipulation in Experiment 1 and the difficulty manipulations in Experiment 2. We also replaced the 2x2 factorial design with a paradigm where we jointly varied stimulus contrast and variability across three levels. Due to these differences, it is difficult to make direct comparisons of accuracy between the networks and humans. Nevertheless, our networks achieved accuracies of 70% that that were comparable to humans in these tasks (humans showed an average of 70% in Experiment 1 and 84% in Experiment 2) and our results show that our task paradigms were indeed able to isolate energy manipulations and replicate their effects. To test the generality of our findings, we also included a novel task paradigm as Experiment 3 that has not been previously tested on humans but nevertheless uses the same kind of energy manipulations.

In Experiment 1, the stimuli (90 x 90 pixels) consisted of an array of eight noisy, oriented Gabor patches. Each individual Gabor patch in the array spanned 30 x 30 pixels. The task was to decide whether the average tilt across the 8 patches was clockwise (CW) or counterclockwise (CCW) from the horizontal (**Fig 1A**). For each image, the average orientation of the Gabor

patches across the eight patches was selected from a Gaussian distribution. The possible range of orientations was [0˚,360˚]. The energy of the stimulus was manipulated across three levels by simultaneously varying two features of the array–the contrast of individual gratings and the variability of orientations across the gratings. While increasing the contrast of the gratings allowed better stimulus visibility and made the task easier, increasing the variability of orientations increased the uncertainty regarding the mean orientation across the patches, thus making the task harder.

In Experiment 2, the stimuli (100 x 100 pixels) consisted of two noisy, sinusoidal gratings (oriented either 45˚ CCW or CW to the vertical) superimposed on each other. The two gratings were always oriented orthogonally to each other and one of the gratings had a higher contrast (referred to as the dominant grating). The task was to determine whether the dominant grating was oriented CCW or CW to the vertical (**Fig 1B**). The energy of the stimulus was manipulated by simultaneously varying the contrast levels of the dominant and the non-dominant grating across three levels. Increasing the contrast of the dominant grating contributed positive evidence making the task easier, while increasing the contrast of the non-dominant grating increased the level of contradictory or "negative evidence" making the task harder.

In Experiment 3, the stimulus consisted of a single noisy Gabor patch (100 x 100 pixels) oriented 45˚ either CCW or CW to the vertical (**Fig 1C**). The task was to identify the direction of tilt (CCW/CW). The energy of the stimulus was manipulated by varying both contrast and noise of the gratings. While increasing contrast makes the task easier, increasing noise degraded the stimulus, making the task harder.

## Generating the training and validation sets

For each experiment, we trained the networks on a set of 10,000 images. To allow the networks to learn generalizable representations of the stimuli, we generated images by sampling the stimulus parameters uniformly within a range. In Experiment 1, we uniformly sampled mean orientation of the gratings from the interval [1˚,10˚], the variability of orientations from the interval [1˚,20˚], and stimulus contrast from the interval [0.01,1]. The possible range of orientations was [0˚,360˚]. In Experiment 2, we sampled the orientation of the gratings from the interval [1˚,45˚], the contrast of the dominant grating from the interval [.01,1] and the difference in contrast between the dominant and non-dominant gratings from the interval [.01,$x$] where $x$ refers to the contrast of the dominant grating which sets an upper bound on the contrast of the non-dominant grating. In Experiment 3, we sampled the contrast of the Gabor patch from the interval [.01,1], and noise (in units of standard deviation) from the interval [.01,2]. Training was validated on a set of 1000 images generated using the same stimulus parameter distributions as the training set.

## Network architectures

We tested three CNN architectures–a 4-layer CNN, VGG-19, and ResNet-50 –on the experiments described above. The networks receive inputs in the form of an image consisting of n x n pixels (n = 90 for Experiment 1 and n = 100 for Experiments 2 and 3) and outputs a binary category label corresponding to the identity of the stimulus (CCW or CW).

The 4-layer CNN model consisted of two convolutional layers (with kernels of size 3 x 3 pixels) paired with two max pooling layers (pooling performed over 2 x 2 pixel windows), one flat layer, and two fully connected layers (consisting of 64 units and 1 unit respectively). A rectified linear unit (ReLu) activation function transformed the outputs of each convolutional layer and the 64-unit fully connected layer, whereas a sigmoid activation function was applied to the output of the final layer.

We also trained two deep CNNs using the standard VGG-19 and ResNet-50 model variants. The VGG-19 model consists of 16 convolutional layers, 3 fully connected layers, 5 max pool layers, and 1 softmax layer. The ResNet-50 model consists of 48 convolutional layers, 1 max pool layer, and 1 average pool layer. The top layer of these networks was modified for binary classification by adding a fully connected layer consisting of a single unit with a sigmoid activation function.

## Training the networks

We trained networks on 10,000 images from each of the three experiments to achieve a classification accuracy > 89% on all tasks. Model performances were assessed on a validation set consisting of 1000 images. The 4-layer CNNs were trained for 25 epochs with a batch size of 32, using the binary cross-entropy loss function and Adam optimizer with a learning rate = 0.001, weight decay = 0 and $\epsilon = 10^{-8}$. As the tasks were relatively simple, to prevent overfitting, we used early stopping with a patience of 10 epochs.

The deep CNNs (VGG-19 and ResNet-50) were trained on these tasks using transfer learning and fine-tuning. We first instantiated the base model pretrained on the ImageNet dataset (provided in Keras Applications at https://keras.io/api/applications/) and froze the model's weights. The classification layer at the top was excluded to enable feature extraction. Next, we added a global average pooling layer to convert the features extracted from each image into a single vector. Finally, we added a classification head with a single unit to convert these features into binary predictions. To prevent overfitting, we also included a drop-out layer with a drop-out rate of 0.2. Using a base learning rate of 0.001 for the Adam optimizer, we trained this model initially on 10 epochs on binary cross-entropy loss. We found that these networks generally showed poor classification performance (~60%), and therefore trained them further by unfreezing and fine-tuning the top layers of the network. For fine-tuning, training was continued for a further 10 epochs. The models were fine-tuned on binary cross-entropy loss using a lower learning rate (0.0001) for the RMSprop optimizer. Fine-tuning improved the models' performances considerably with all models now achieving a classification accuracy of at least 89%.

For each type of network and each experiment, we determined the optimal number of layers to fine-tune by incrementing the number of fine-tuning layers in steps and assessing model performance. We chose the model that gave us the highest accuracy while minimizing the number of layers to fine-tune. For VGG-19, the best models consisted of 8 fine-tuned layers for Experiments 1 and 2 and 5 fine-tuned layers for Experiment 3. For ResNet-50, the best models consisted of 40 fine-tuned layers for Experiments 1 and 2 and 10 fine-tuned layers for Experiment 3.

Finally, to allow for individual differences in learning, we trained 25 instances of each of the three models separately for each experiment using a different random seed to initialize the network's weights before training.

## Determining stimulus parameters for energy manipulations

To induce confidence-accuracy dissociations, we need to jointly manipulate the signal strength and variability/negative evidence ("energy") of the stimulus such that the network's accuracies are matched across conditions. Therefore, we need to determine the stimulus parameters that will allow us to obtain matched network performances across the three conditions. To do so, we first performed a coarse search by fixing the stimulus along the "contrast" dimension for each energy condition (contrast of the Gabor patches for Experiments 1 and 3 and contrast of the dominant grating for Experiment 2) and varying it along the "noise" dimension (variability

of orientations for Experiment 1, contrast of the non-dominant grating for Experiment 2 and noise in Experiment 3) in relatively large steps. Next, for each energy level, we determined a range of noise values that gave us a target accuracy between 65–75% and performed a fine-grained search within this range for the parameters that resulted in an accuracy of 70%.

The search yielded stimulus parameter estimates that resulted in matched accuracy levels of 70% across the three energy conditions for each of the three types of networks. Specifically, we obtained the following parameters for the 4-layer CNNs (Experiment 1: contrast = [.2, .25, .3], orientation variance = [7.35˚, 21.28˚, 27.28˚]; Experiment 2: dominant contrast = [0.2, 0.4, 0.6], non-dominant contrast = [0.168, 0.375, 0.575]; Experiment 3: contrast = [.05, .1, .15], noise = [.42, .82, 1.21]), VGG-19 (Experiment 1: contrast = [.4, .5, .6] and orientation variance = [21.42˚, 25.28˚, 27˚]; Experiment 2: dominant contrast = [0.2, 0.4, 0.6] and non-dominant contrast = [0.13, 0.358, 0.56]; Experiment 3: contrast = [.05, .1, .15] and noise = [.32, .54, .715]), and ResNet-50 (Experiment 1: contrast = [.4, .5, .6] and orientation variance = [17˚,18.5˚, 20˚]; Experiment 2: dominant contrast = [0.2, 0.4, 0.6] and non-dominant contrast = [0.13, 0.358, 0.555]; Experiment 3: contrast = [.05, .1, .15] and noise = [.29, .46, .607]). Using these parameters, for each of the three energy levels, we generated stimulus sets consisting of 1000 images to test the CNNs for confidence-accuracy dissociations.

## Behavioral analyses

**Accuracy and confidence of the networks.** The final layer of the network consists of a single unit whose activation ($a$) arises from a sigmoid activation function. The network's responses ($r$) were generated such that, $r = \begin{cases} S_1, if\ a < 0.5 \\ S_2, if\ a \geq 0.5 \end{cases}$ and decision confidence ($c$) was generated as, $c = \begin{cases} 1 - a, if\ a < 0.5 \\ a, if\ a \geq 0.5 \end{cases}$ where $a \in [0,1]$.

We computed the average accuracy and confidence separately for each of the 25 network instances and for each energy condition.

**Measures of type-1 and type-2 sensitivity.** An observer's type-1 or perceptual sensitivity (d') is a measure derived from signal detection theory (SDT) which quantifies the observer's ability to discriminate between the two stimulus categories [17]. Type-1 sensitivity (d') is defined as, $d' = \phi^{-1}(HR) - \phi^{-1}(FAR)$ where HR and FAR refer to the observed hit rate and false alarm rates, respectively, when the stimulus category $S_2$ is treated as the target, and $\phi^{-1}$ is the inverse of the cumulative standard normal distribution that transforms cumulative probabilities into z-scores.

Type-2 or metacognitive sensitivity (meta-d') is a measure derived from SDT-modelling of the observer's decision and confidence responses which quantifies the information underlying the metacognitive judgement [68]. Intuitively, it can be thought of as a measure of the observer's ability to distinguish between their own correct and incorrect responses using confidence responses.

## Assessing the networks' final layer activations

To understand the effect of energy manipulations on the distributions of evidence within the network, we studied how the distribution of the network's activations change with energy levels. Each time an image is presented to the network, it produces an activation in the output layer. We aggregated these activations across all instances of the network separately for all images from each energy condition. We then visualized the distributions of these activations separately for images from each stimulus category using kernel density plots.

To quantify changes in the characteristics of these distributions, we computed two measures–the difference in means of the $S_1$ and $S_2$ distributions (which quantifies the separation in evidence between the two categories) and standard deviation of these distributions averaged across the two distributions (which quantifies the degree of uncertainty associated with identity of the stimulus). We computed these measures separately for each network instance and energy condition, and averaged across network instances.

## Simulating the task paradigm for generating meta-d'/d' dissociations

It has been previously demonstrated that a certain task paradigm can induce dissociations between observers' meta-d' and d' [18,28]. Particularly, in a two-choice task involving discrimination between two stimulus categories ($S_1$ and $S_2$), when the contrast of one stimulus category ($S_1$) is held fixed while the contrast of the other category is allowed to vary across trials ($S_2$), meta-d' is found to increase with d' on trials where the observer responds "$S_2$" and found to decrease with d' on trials where the observer responds "$S_1$."

We simulated this paradigm for Experiments 1–3 by fixing the stimulus contrast for one of the stimulus categories (CCW) and allowing the contrast of the stimuli from the other category (CW) to vary discretely across five levels. Specifically, in Experiment 1, the CCW stimulus was fixed at .225 and the CW-tilted stimuli was varied along the range [.05, .135, .225, .3125, .4]. In Experiment 2, the CCW was fixed at .21 and the CW tilted stimuli was varied along the range [.2, .205, .21, .215, .22]. Finally, in Experiment 3, the CCW was fixed at .1 and the CW tilted stimuli was varied along the range [.05, .075, .1, .125, .15]. The contrast levels of the stimuli were chosen via simulations such that the network's perceptual sensitivity (d') spanned a range of meaningful values (1 to 3.5) while avoiding floor or ceiling effects. We generated test sets of 1000 images for each contrast level assumed by $S_2$ and obtained the decision and confidence responses from our previously trained networks.

We computed d' and meta-d' separately for each contrast level of $S_2$. For response-specific assessment of meta-d', we computed meta-d' separately for trials conditioned on each type of network response ($S_1$ vs $S_2$ responses) and contrast level.

## Color discrimination experiment

The task paradigm for the color discrimination experiment was adapted from Boldt et al., (2017) [11]. This task was previously shown to produce confidence-accuracy dissociations in humans where confidence decreased with stimulus energy levels [9–12,14], in contrast with previous findings where confidence increased with energy levels.

The stimuli (90 x 90 pixels) consisted of an array of eight colored circular patches. Each individual color patch in the array spanned 30 x 30 pixels. The task was to decide whether the average color across the 8 patches was more blue or red (**Fig 6A**). For each image, the mean color across the eight patches was selected from a uniform distribution with a mean color intensity of $c$ and an interval of width $v$. The stimulus parameter $c \in [0,1]$ controlled the intensity of "redness" or "blueness" along a continuous range such that $c = 0$ yielded a completely red patch and $c = 1$ yielded a completely blue patch and values in-between resulted in patches containing a mixture of red and blue with varying proportions of each color. In general, a patch with $c < .5$, contained more red and a patch with $c > .5$ contained more blue. The parameter $v$ controlled the variance of color in these patches with higher values of $v$ resulting in more variable colors within an array. The energy of the stimulus was manipulated across three levels by simultaneously varying these two features of the array–the color intensity and the spread of color intensity. While increasing the color intensity made it easy to identify the "blueness" or

"redness" of the color and made the task easier, increasing the color variance increased the uncertainty regarding the mean color across the patches and made the task harder.

As in our main analyses, we trained 25 instances of the three networks (4-layer CNN, VGG-19 and ResNet-50) using the same procedure outlined above. We determined the stimulus parameters that would allow us to obtain matched network performances across the three energy conditions. The search yielded stimulus parameter estimates that resulted in matched accuracy levels of 70% across the three energy conditions for each of the three types of networks– 4-layer CNNs (color intensity = [.493, .492, .49], color variance = [.4, .494, .626]), VGG-19 and ResNet-50 (contrast = [.4, .3, .2], color variance = [.85, .95, .97]). Using these parameters, for each of the three energy levels, we generated stimulus sets consisting of 1000 images and tested the accuracy and confidence of each of the 25 instances on these images. Finally, as before, we examined the separability and spread of the distributions of evidence.

## Supporting information

**S1 Fig. Assessing resolution and calibration of confidence.** (A) We tested whether the CNN's confidence could reliably discriminate between correct and incorrect choices for the different energy conditions. We found that for all stimulus energy levels and all three networks (4-layer CNN, VGG-19 and ResNet-50) confidence was higher for correct choices compared to incorrect choices, confirming that the networks' confidence reliably tracked their accuracies. (B) CNNs tend to be overconfident and generate confidence values greater than their accuracies. Therefore, we tried to calibrate the CNNs' confidence using a post-processing method called "temperature-scaling" [29] which uses a parameter ($T$) that can scale the network's confidence to match its accuracy. The network's output in logits ($z$) is scaled before the sigmoid transformation such that $z_{scaled} = \frac{z}{T}$. Confidence is then computed as $\frac{1}{1+e^{-\frac{z}{T}}}$ and therefore, when $T > 1$, the network's confidence gets scaled down. We found that temperature scaling indeed reduced overconfidence in all networks (4-layer CNN: confidence decreased from 0.968 to 0.737; VGG-10: confidence decreased from 0.964 to 0.783, ResNet-50: confidence decreased from 0.982 to 0.748). However, for all networks we still robustly observed the confidence-accuracy dissociations.
(TIFF)

**S2 Fig. The signal-and-variance-increase hypothesis can explain the effect of base-rate changes on choice behavior during energy manipulations.** Herce Castañón et al. (2019) [13] showed that when subjects are asked to judge the mean orientation of an array of Gabor patches, they exhibit a range of suboptimal behaviors. Specifically, Herce Castañón et al. [13] tested how energy manipulation interact with changes in stimulus base rates by measuring the shifts in subjects' decision criteria in response to changes in stimulus frequencies for each energy condition. The location of the decision criteria was computed from the SDT-based measure of response bias ($c$). The shift in criteria in response to changes in stimulus base rates was quantified as $c_{S_1} - c_{S_2}$ where $c_{S_i}$ refers to the criterion in the condition where stimulus $i$ is more frequent. Herce Castañón et al. [13] found that subjects showed significantly smaller shifts in their criterion in response to base rate changes in the high-energy condition, compared to the low-energy condition. They explained these effects by proposing that observers are blind to the noise arising from their own cognitive computations, thus resulting in failure to account for the higher levels of uncertainty arising from high-energy stimuli. Here, we show via simulations that the signal-and-variance-increase hypothesis can also explain their observed effects. We simulated the effect of stimulus energy manipulations in an SDT model where higher stimulus energy led to increase in signal ($\mu$) as well as variance ($\sigma$) of the stimulus

evidence distributions. We generated 50 simulations by sampling individual SDT parameters: the signal, $\mu$, and the decision criterion, $c$, from Gaussian distributions. Specifically, in the low-energy condition $\mu_{low} \sim N(1,.5)$ and in the high-energy condition $\mu_{high} = N(2,1)$, such that the two distributions would lead to equal sensitivity but the high-energy condition features higher variance of the evidence distributions. Similarly, the decision criterion for individual simulations was sampled from $c \sim N(0,.25)$. For the base-rate manipulations, we assumed that an increase in probability of observing the stimulus class $S_1$ would shift the criterion by an amount $+c_{shift}$ to allow more frequent $S_1$ responses. Similarly, increase in probability of $S_2$ stimuli would lead to a criterion shift of $-c_{shift}$. We allowed individual variability in criterion shifts by sampling $c_{shift} \sim N(.4,.2)$. Critically, we assumed that these criteria were fixed across the two energy conditions. Using these parameters, we generated data for 10,000 trials from each of the 50 simulations and computed the SDT-derived measures of stimulus sensitivity ($d'_{obs}$) and response bias ($c_{obs}$) from these data separately for each energy condition and base-rate condition. As done by Herce Castañón et al. (2019) [13], we computed the bias index as the difference in $c_{obs}$ between the two base-rate conditions. The plots show the average $d'_{obs}$ and bias index across all simulations, separately for the low- and high-energy conditions. Our findings replicate the original observations that in spite of matched performance between the two energy conditions ($\Delta d' = .0024, t(49) = -.684, p = .50$), observers appear to shift their decision criterion less in favor of the more probable stimulus in the high-energy condition ($\Delta$bias index $= .39, t(49) = 13.66, p < .0001$). These findings show that the signal-and-variance-increase hypothesis can indeed capture the suboptimal behaviors attributed to the noise-blindness mechanism. ***p<0.001; n.s., not significant.
(TIF)

**S1 Table. Comparisons of mean confidence, separability and variance of evidence distributions between the low and high stimulus energy conditions for the three networks (4-layer CNN, VGG-19 and ResNet-50) across the four experiments.** Stimulus energy manipulations consists of simultaneous manipulations of contrast and variability in the same direction. Comparisons using two-sided, pairwise t-tests showed that confidence as well as separability and variance of the evidence distributions increased significantly from low to high stimulus energy condition for all networks and all experiments.
(XLSX)

**S2 Table. Comparisons of separability and variance of evidence distributions when noise and variability are manipulated independently for the 4-layer CNN across the three experiments.** For contrast manipulations, the stimulus contrast was increased while stimulus variability was held constant. For variability manipulations, the stimulus noise or variability was increased while stimulus contrast was held constant. Comparisons using two-sided, pairwise t-tests showed that separability of evidence distributions increased significantly from the low- to high-contrast condition whereas the variance of evidence distribution decreased from the low- to high-contrast conditions for all experiments except Experiment 3. For Experiment 3, increasing contrast increased the variance of evidence distributions. On the other hand, separability of evidence distributions decreased significantly from the low to high variability condition and the variance of evidence distribution increased from the low to high variability conditions for all experiments.
(XLSX)

## Author Contributions

**Conceptualization:** Medha Shekhar, Dobromir Rahnev.

**Data curation:** Medha Shekhar.

**Funding acquisition:** Medha Shekhar, Dobromir Rahnev.

**Investigation:** Medha Shekhar, Dobromir Rahnev.

**Methodology:** Medha Shekhar, Dobromir Rahnev.

**Project administration:** Medha Shekhar, Dobromir Rahnev.

**Resources:** Dobromir Rahnev.

**Software:** Medha Shekhar.

**Supervision:** Dobromir Rahnev.

**Validation:** Dobromir Rahnev.

**Writing – original draft:** Medha Shekhar.

**Writing – review & editing:** Medha Shekhar, Dobromir Rahnev.

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
