## [Decision Letter · Decision Letter 0]

20 Jun 2024

Dear Dr. Shekhar,

Thank you very much for submitting your manuscript "Human-like dissociations between confidence and accuracy in convolutional neural networks" for consideration at PLOS Computational Biology.

As with all papers reviewed by the journal, your manuscript was reviewed by members of the editorial board and by several independent reviewers. In light of the reviews (below this email), we would like to invite the resubmission of a significantly-revised version that takes into account the reviewers' comments. 

As you will see (and I), both reviewers found much merit in your paper. However, both reviewers raise some very valid concerns. I think that addressing these concerns will make the paper much stronger and its potential impact much greater. 

We cannot make any decision about publication until we have seen the revised manuscript and your response to the reviewers' comments. Your revised manuscript is also likely to be sent to reviewers for further evaluation.

Sincerely,

Leonidas Doumas

Guest Editor

PLOS Computational Biology

Andrea E. Martin

Section Editor

PLOS Computational Biology

Reviewer's Responses to Questions

**Comments to the Authors:**

Reviewer #1: Review “Human-like dissociations between confidence and accuracy in convolutional neural networks”

The current work presents a series of simulations testing the hypothesis that dissociations between confidence and accuracy found in the psychological literature can be explained by low level changes in the separation and variance of perceptual representations. The overall strategy consisted of training deep neural networks (DNNs) to classify Gabor patches stimuli into clockwise and counterclockwise categories and testing on stimuli groups varying on energy (contrast plus variability) in three main experiments. The results showed that three different models (a custom 4-layer CNN, a VGG-19, and a ResNet-50), showed a dissociation between accuracy and confidence, with accuracy being constant while confidence increased with energy.

I think this is an important work in perceptual decision-making as it helps to disentangle a current debate through simulations with image computable DNNs. I especially liked the rigorous experimental approach taken by the authors (see comments below). However, I do have some concerns that need to be addressed before I can recommend this manuscript for publication, which I detail in the following:

Section “Different stimulus features selectively influence the separability and spread of internal evidence distributions”:

• How were the main 3 experiments conducted? Before this section it seemed that Energy was a single independent variable in the experimental design of the 3 main studies, but in this section, it is revealed that, in fact, Contrast and Variability were also factors. Were these groups added to the groups of experiments 1-3 for comparison or experiments 1-3 had always Contrast and Variability as independent variables that were combined to generate the high energy condition? From the writing this is unclear.

Section “CNNs can reproduce dissociations between type-1 and type-2 sensitivity typically regarded as evidence for the positive evidence mechanism”:

• Why was this study only run with the custom 4-layer network? Since the results are based on the activation of the output (single unit with sigmoid activation), it should be possible to run the study with the VGG-19 and ResNet-50 models, especially considering that the main result, the cross-over between the S1 and S2 responses in their meta-d’ and d’ relationship, was not found for Experiment 2. I strongly encourage the authors to run with the VGG-19 and ResNet-50 models to check the robustness of their results in this section.

Section “CNNs as models for understanding human vision”:

• While it is true, that many have argued that DNNs can provide insights into the workings of the visual system, many of the papers cited by the authors in this regard fail to emphasize what I think is one of the strengths of the current work: the manipulation of independent variables in order to contrast theoretically motivated hypothesis. Bowers et al. (2023a) have empathized the prevalent lack of experimental manipulations on deep learning research. Furthermore, Bowers et al. (2023b) have argued for the need of severe testing of DNNs as models of human cognition. I think the discussion of the current work would benefit from making its methodological strengths in comparison to correlational DNNs studies more explicit.

References:

Bowers, J. S., Malhotra, G., Adolfi, F., Dujmović, M., Montero, M. L., Biscione, V., ... & Heaton, R. F. (2023a). On the importance of severely testing deep learning models of cognition. Cognitive Systems Research, 82, 101158.

Bowers, J. S., Malhotra, G., Dujmović, M., Montero, M. L., Tsvetkov, C., Biscione, V., ... & Blything, R. (2023b). Deep problems with neural network models of human vision. Behavioral and Brain Sciences, 46, e385.

Reviewer #2: The authors investigate whether CNN architectures can help to distinguish between two different possible mechanisms that might explain why stimulus energy (e.g., contrast, variability) impacts confidence ratings (but not accuracy) in a visual discrimination task. One hypothetical mechanism is a high-level ‘positive evidence heuristic’ account and another is framed as the result of low-level visual processing. Because CNNs do not have a mechanism for confirmatory bias, the authors think that these networks can be useful for deciding between the two possibilities.

The work is likely of interest to people using signal detection modeling to try to understand visual discrimination behavior. The main contribution of the paper is an existence proof using CNNs that a confirmatory bias heuristic is not required to explain the dissociation between confidence and accuracy in visual processing. This seems to be related to the Webb et al. (2023) paper that the authors cite.

However, the paper is framed as a project that directly investigates the contributions of high and low level processing to this dissociation effect in people, using CNNs as models of the human visual system. Here the paper is not successful in its aims. There are several problems with this framing of the work that mostly stem from the choice of CNNs as a model. While the existence proof that the prior SDT modeling research was flawed in its conclusions is a nice contribution, that does not mean that all types of high level processing are eliminated as explanations for, or contributions to, the effect in question in the human visual system.

The authors show in the paper that manipulating stimulus energy at the input to a CNN causes changes in the distribution of patterns on the output layer in ways that are consistent with the dissociation effect between confidence and accuracy. However, even if we were to take CNNs at face value as models of human vision (which we definitely should not for reasons elaborated below), this work is not able to speak to the level of processing in any general sense. CNNs, including the ones used in this research, are ambiguous in terms of where representations lie in the visuo-cognitive processing stream. The input to a CNN is probably something like LGN. The output of a typical CNN is often an object label, which should correspond to late vision at the earliest, or according to some accounts, language or other cognitive processes that probably involve prefrontal cortex. So demonstrating this effect using a CNN does not isolate the mechanism to “low level” processing in any way. This claim is an overinterpretation of their results.

Another issue with the CNN architecture is that CNNs are strictly feed forward, and the human visual system is not, with large numbers of back projections from prefrontal cortex and lateral connections at all levels of representation. There is broad evidence in the literature of continuous interactions of top-down, bottom-up, and lateral effects. By virtue of their architecture, CNNs cannot capture the fact that almost nothing in human vision is strictly bottom-up or top-down. What is cognitively penetrable at any level of representation also varies. The dichotomy between low-level and high-level processing that the paper (and perhaps the SDT field?) sets up seems to be behind what is known in the visual cognition literature.

There are also some issues with the way a CNN and its representations are described in the paper, with some confusion about what a network’s ‘representations’ are (as opposed to mathematical or statistical characterizations of input or output data). For example, the authors refer to patterns of activity that they have characterized on the output layer of the CNN using a signal detection framework as ‘internal representations’. I think this description is misleading because CNNs do have actual internal representations (even if they are incomprehensible).

If the paper could be reframed to speak to the issues in SDT modeling, and resist over claims about CNNs being good models of human visual cognition, it would be stronger. Late in the paper, in the discussion, the core issues about modeling the phenomenon are more clear.

Some of the statistics reporting needs to be cleaned up. A few sections could use editing for clarity or readability.

Please see specific notes below by page:

Page 6 (abstract)

The abstract could use some more context. Going in cold, I did not know from the abstract or title that this work was related to SDT modeling.

Page 7

Paragraph 2:

Remove the ‘s’ from dissociations.

Not sure what "strength of discomfirming evidence" means in this context, and there are no examples/citations.

In general, paragraph 2 could use some finessing for clarity and readability.

Page 8

Paragraph 1

This paragraph could be finessed for clarity and readability

Does the term "positive evidence heuristic" meaningfully differ from confirmation bias? I did not find this term widely used in the literature or in the given citations.

The description of noise blindness and how it works is not clear.

Paragraph 2

It is not clear why energy manipulations would “naturally” lead to higher confidence

The authors should be careful to distinguish signal detection theory (which is something that can characterize data) from actual mental representations. Signal detection is quite limited in terms of providing representational or mechanistic explanations.

Greater input contrast to any neuron that is filtering for a pattern at any level of hierarchy

in a network will probably cause greater differentiation on the output. It's not just limited to low level filtering. In a network like a CNN the increased contrast would likely propagate through the whole network.

Page 9

I am not at this point convinced that a signal detection model sheds any light at all on whether this effect is low level or high level (or a third alternative that it is some combination thereof). There needs to be much better support for the claim that a signal detection characterization corresponds solely to low-level processing. Why would this be the case? What is the proposed relationship to low-level mechanisms?

Are these two "explanations" perhaps not explanations at all? One is a well-known effect and the other is a computational theory-level mathematical characterization (at best). It's true that neither provides a specific mechanistic explanation, however.

Paragraph 3: There are other possible reasons that a CNN would fail to mimic human behavior other than a lack of a built in positive evidence heuristic. For example, CNNs don't work like the human visual system.

Page 10

This section could use some finessing for clarity and style/vocabulary.

'anticipate' is probably the wrong word here.

“In addition, they reproduced another signature of confidence that is popularly regarded as

evidence for the positive evidence bias (Maniscalco et al., 2016)” What signature is this?

“Importantly, these observations highlight the common mechanisms underlying the behavior of humans and artificial systems, suggesting that these confidence accuracy dissociations may be driven by external features of the environment, without the need for specialized internal cognitive mechanisms.” It seems like it would be very difficult to make this claim using a CNN, again given that we know CNNs are not really anything like human vision. A more modest claim is made at the end of the paper, which is that 1) prior SDT research may have reached inappropriate conclusions and 2) a CNN can act as an existence proof that a network *may* not need high level processing (but that what humans do is another question).

Page 11

Nowhere in this paper am I really getting a feel for how well the models really mimic human data for these tasks. I think this is important for evaluating claims about humanlike visual discrimination. For example, is the performance level of 70% an arbitrary decision, or does it correspond to something in the prior experiments?

Page 12

Figure 1 Caption

Clockwise and counterclockwise doesn't specify orientation (any orientation can be reached via clockwise or counterclockwise rotation--even from a 12:00 (vertical) reference point).

Paragraph 1: By ‘average’ to you mean the mean? Or another measure of central tendency?

Page 13

Figure 2

It's very hard to compare what the different models are doing when they are not plotted on the same scale.

Page 14

Strike 'highly significant' and just use ‘significant’. It's not a measure of the importance of the result.

Again, I am looking for more detail about the match to the human data given the claims that the network is matching that data. Can you reproduce their figures/plot their data to make this comparison more clear?

“These findings cast doubt on the necessity of high-level explanations…”

I agree that doubt is cast on prior accounts, but I disagree with the general theme of the paper that that these results differentiate between high and low level processing in people. The output nodes in CNNs typically correspond to some sort of high-level object labeling, which a lot of researchers take to model high level vision at least, and maybe even language and cognition. True, there is no explicit built in confirmation bias mechanism in a CNN, but I still have reservations that a CNN is even a good approximation of the human visual system in the first place. In the human visual system, backprojections abound, for example. Vision is not feed-forward and low level processing is not truly independent of high level processing (or lateral interactions).

Page 15

“While findings of confidence-accuracy dissociations in CNNs support the signal-and variance-increase hypothesis” I think this is probably too strong a claim if the idea is that it applies to people.

“We accessed the CNNs’ internal representations by aggregating the activations generated in the networks’ output layer…” The output layer is not low level in a CNN, though. How is this an internal evidence representation having anything to do with low level processing? Furthermore, even if you were actually looking at internal representations (hidden layers), you can't really point to any node in a network with distributed representations like a CNN (with the exception of the inputs or outputs) and say that it corresponds to any specific representation, so I am skeptical that any specific representation in these networks could be probed at all. CNNs just don’t help us understand people.

Page 16

Strike ‘highly significant’ and just use ‘significant’ for the reasons explained above.

Report stats APA style with details of the test and statistic values.

Page 17

I still feel that I need to better understand the human data given the number of comparisons in the paper. We have a not-great model of human vision that is supposed to be simulating human data, but we don't really know whether it does that well in the first place, from what is reported here.

Here the scales are also not the same between plots.

While one would probably expect that showing the network greater contrast would cause the separation reported here at the output of the CNN, it is not made explicit how this relates to human behavior or representation.

Page 18

Use APA style. What is the exact statistical test, over what exact data?

“The concurrent increase in separation between the two stimulus categories and the

variability of evidence ensures that the network’s overall stimulus sensitivity remains

constant between the three energy conditions…” This sounds like the signal detection model is having an effect on the CNN, so the sentence should be changed for clarity. (The point is clarified in the next sentence.)

“These findings show that the signal-and-variance-increase mechanism indeed underlies the confidence-accuracy dissociations produced by the CNNs.” This is misleading because there is no "signal and variance increase mechanism" in a CNN. The SDT characterization of the output nodes changes, but that is not a mechanism in the network.

Page 19

Earlier in this paper 'energy' was defined according to characteristics like contrast and variability, but now contrast and variability are being compared to energy. The manipulation needs to be clarified.

Once again it seems strange to use the term 'internal evidence distributions' given that it's characterized over the output.

The reporting of the t-test should include what exact data was compared and use APA style with t statistics.

Page 20 - 21

Completely report statistical tests using APA style

“Overall, these results demonstrate that manipulations of stimulus contrast and variability

have opposite effects on the separability and spread of internal activations…” I can believe this claim, but I am not sure how it relates to the question this paper aims to investigate. This does characterize CNN responses to changes in input characteristics, but it doesn't really speak to mechanism or level of representation in people.

Page 22

“So far, our results have shown that CNNs, in spite of lacking the positive evidence

mechanism, can produce human-like confidence-accuracy dissociations…” given the lack of detail about the human data this is probably too strong a claim about the network producing ‘human-like’ results.

Consider reordering the sentences about type 1 and type 2. I wasn't sure which kind of type-1 and type-2 the authors were referring to at first.

Page 23

“Overall, our findings demonstrate that for at least for some stimulus manipulations, low-level mechanisms are sufficient to explain effects that have typically been taken as evidence for a positive evidence mechanism.” This is an interesting result with respect to the prior claims using SDT models as evidence for the necessity of a certain kind of high-level processing. This is relevant to researchers using SDT models. I am still unconvinced that the alternative account should be called ‘low-level mechanisms’ given what we know about human visual processing, and the fact that CNNs don’t reveal much about how they represent information.

Page 24

“Our previous findings establish that low-level changes in perceptual representations can

explain human behavior that has typically been attributed to high-level cognitive

mechanisms. Here, we sought to further test whether the “robust averaging” mechanism

can also be realized through low-level mechanisms” The previous findings established that changes in inputs to CNNs produce patterns in a signal detection framework that have previously been attributed in other signal detection modeling to high level cognitive mechanisms. While this casts doubt on prior SDT work, this a reminder that there is not a real connection to the human visual system’s representations yet. There are no actual mechanisms specified by this SDT framework.

Page 25

Strike ‘highly significant’ and replace with ‘significant’ for reasons previously stated.

“Nevertheless, these results while establishing the generalizability of the mechanisms underlying confidence-accuracy dissociations in CNNs also reveal how stimulus-specific interactions can constrain the similarities between humans and CNNs.” This sentence should be reworded for clarity. I’m not really sure what the claim is.

Page 27

“Our findings support the alternative, signal-and-variance increase hypothesis which posits that such dissociations naturally emerge from low-level changes in perceptual representations.” This is too strong a claim. The authors’ findings support the idea of a relationship between the energy characteristics on the input layer of a CNN and the distribution of activity on the output layer, and that prior SDT research might have come to incorrect conclusions about certain types of high level accounts of this effect.

“…could be driven by common, stimulus-driven processes and demonstrate the usefulness of CNNs in distinguishing between low- and high-level explanations of behavior.” This statement is not really supported by this work, again because CNNs are very different than the human visual system in terms of their architecture particularly recurrency and backprojection. CNNs by virture of their simplistic architecture can’t speak to high level and low level interactions. The link to what is actually known about human visual representation is missing.

Page 28

This is a great passage and in my opinion is really what this paper is about: evidence against prior SDT conclusions. The prior PE literature perhaps had some flaws that led to poor conclusions about human visual discrimination and representation, in part because data characterized by SDT models might have many different kinds of underlying mechanisms, and SDT does not generally specify a mechanism.

Page 30

“..observers’ failure to scale their decision and confidence criteria in response to low-level

perceptual changes” Be careful here. Some low level changes are not cognitively penetrable, but others are. It's not clear that anything can be said about decision criteria on low-level stimuli subject to noise. Again, there is constant interaction between top-down and bottom-up processing in the real visual system.

Unless you're analyzing the internal layers of the CNN, it's still quite strange to refer to 'internal representations'

The TMS data is interesting, but it’s coming quite late in the paper, and the same issues with CNNs apply. The link between the TMS, the SDT, and the CNN is tenuous.

“Several recent studies have argued that deep neural networks can provide meaningful

insights into the goals and constraints that have shaped human perception..” Bowers et al. (2023) argued that they cannot. Bowers et al. (2023). Deep problems with neural network models of human vision. Behavioral and Brain Sciences, 46, e385

“Indeed, our findings support this argument as they carry implications about the external

constraints that may have shaped visual processing.” This is too much of an overclaim. CNN's cannot capture the majority of human behavior in vision. Again, see Bowers et al. (2023)

“Another critical advantage of CNNs is that they can allow us isolate bottom-up visual processes from the top-down mechanisms that serve cognition, since visual processing in standard CNNs occurs in the absence of specialized top-down influences.” Again, this is an overclaim. In this paper the authors have demonstrated that CNNs can show that there is a network that exists that can produce the behavior that prior researchers claimed was only possible with specific top-down influences.

“In addition, they can inform us about the possible stimulus and task representations that

underlie such bottom-up stimulus processing (Green et al., 2024).” This sentence seems disconnected from the rest of the paper.

Page 33

“These findings cast doubt on the necessity of invoking high-level, cognitive explanations for this phenomenon – particularly the popular assumption that confidence is derived from a positive-evidence heuristic.” This is a much more measured and accurate claim that is still scientifically important, especially to researchers using SDT.

“Our results highlight the necessity of disentangling low- and high-level explanations of behavior and establish CNNs as promising models for generating and testing hypotheses about the mechanisms underlying human behavior.” This work does not really establish the usefulness of CNNs for understanding human vision. It does establish the usefulness of CNNs for disproving prior claims about human vision in SDT modeling.

Page 34

“The task was to decide whether the average tilt across the 8 patches was clockwise (CW) or counterclockwise (CCW) from the horizontal (Figure 1A).” What is the range of possible orientations? In principle could it wrap all the way around? Probably not but what’s the limit?

Page 35

Same as page 34 with range of possible orientations.

Reviewer #3: Review for PCOMPBIOL-D-24-00636

The positive evidence bias for confidence (PE) refers to the intriguing observation that confidence in the accuracy of a decision appears to depend more on evidence supporting the chosen option than on evidence supporting non-chosen alternatives. Despite extensive research, the cause of PE bias remains unknown. The manuscript entitled "Human-like Dissociations Between Confidence and Accuracy in Convolutional Neural Networks" examines the emergence of PE bias in CNNs under manipulations of stimulus energy. The study questions the need for "high-level" cognitive mechanisms to explain PE bias, and presents evidence for the signal-and-variance-increase hypothesis through extensive experimentation with different CNN architectures. I found the paper to be well written and interesting, and I recommend publication pending the authors addressing the concerns outlined below.

Comments:

1.

An important contribution of the paper is that, unlike the study by Webb et al. in 2023, confidence is not explicitly trained, but rather read-out from a network trained to perform different classification tasks. It would be beneficial to clarify early in the results section how confidence is computed from the CNN output, since this is pivotal to the paper and sets it apart from previous studies. That is, explain that a final layer with one node (activated by sigmoid) is added to the networks, and confidence is computed as a rectification of this activation. 

 2.

The validity of the study rests on the assumption that the confidence readout from the CNN accurately models human confidence judgments. However, some aspects of the data suggest that this is not a valid assumption. In many of the reported experiments, confidence levels approach ceiling values, suggesting that they may have little discriminative power with respect to decision accuracy. For example, in Experiment 1 (Figure 2), the mean confidence exceeds 0.95 across all energy levels, suggesting that confidence has minimal ability to discriminate between correct and incorrect decisions.

The findings would be more meaningful if the authors could identify operational settings for these networks where confidence resolution and calibration more closely mirror those observed in human experiments for similar tasks.

3.

I found the distinction drawn between "high-level" and "low-level" explanations of PE bias not particularly helpful. The authors suggest that CNN outputs are considered "low-level," while additional processes are deemed "high-level." They argue for a low-level explanation of PE bias based on their findings, but I remain uncertain about this conclusion.

The authors define confidence as a transformation of activation in the CNN's final layer. This transformation might be viewed as "high-level" because it involves an additional operation on the network's output rather than being directly performed by the neural network itself.

In the studt, confidence is assumed to be the distance between “a” and the decision threshold. Because this transformation is unaffected by stimulus noise, it is, in my view, identical to models of confidence that assume "noise insensitivity," a concept the authors regard as "high-level." Had the authors employed a transformation sensitive to noise, the observed PE bias might not have emerged. If so, then the critical factor for observing the PE bias is how confidence is read-out from the activation “a”.

My specific suggestions here are:

(i) Be more explicit about what is meant by “low level” and “high level”, such that it is clear why the signal-and-variance-increase hypothesis is considered “low level” and the noise blindness hypothesis “high level”. I think the paper would be better without framing it in terms of “low” vs “high” levels, but of course that is up to the authors.

(ii) Explore if a mapping from activation “a” to confidence that does take noise into account eliminates the PE bias. If so, the PE bias still depends on the readout mechanism, as postulated by “high level” theories.

4.

One issue with showing the “violin” plots with individual data points is that highly statistically significant differences may not be visually apparent (e.g., the confidence comparison for the VGG-19 network in Figure 2). To improve clarity, authors might consider keeping the "violin" plots to illustrate the distribution of the data, but show means with error bars instead of individual data points.

**Have the authors made all data and (if applicable) computational code underlying the findings in their manuscript fully available?**

Reviewer #1: Yes

Reviewer #2: Yes

Reviewer #3: Yes

PLOS authors have the option to publish the peer review history of their article (what does this mean?). If published, this will include your full peer review and any attached files.

Reviewer #1: No

Reviewer #2: No

Reviewer #3: No
---

## [Decision Letter · Decision Letter 1]

25 Sep 2024

Dear Dr. Shekhar,

Thank you very much for submitting your manuscript "Human-like dissociations between confidence and accuracy in convolutional neural networks" for consideration at PLOS Computational Biology. As with all papers reviewed by the journal, your manuscript was reviewed by members of the editorial board and by several independent reviewers. The reviewers appreciated the attention to an important topic. Based on the reviews, we are likely to accept this manuscript for publication, providing that you modify the manuscript according to the review recommendations.

You will note that all reviewers comment on the improvement of the current version of the manuscript and thank you for taking their concerns to heart in the revision. One reviewer makes some final notes on corrections, but also on the important point about the clarity of using the term "internal" to refer to non-hidden layers. I agree that this point is confusing and does not follow with traditional use in the literature. I think that making these final corrections will be valuable to the clarity and potential impact of the manuscript. I will likely not need to send the paper out for any additional reviews. 

Sincerely,

Alex Leonidas Doumas

Academic Editor

PLOS Computational Biology

Andrea E. Martin

Section Editor

PLOS Computational Biology

Reviewer's Responses to Questions

**Comments to the Authors:**

Reviewer #1: The revised version of this manuscript addresses all my previous concerns, therefore I recommend to accept.

Reviewer #2: I appreciate the author's responses and substantial revisions, which address many of the issues in the original version of the manuscript. The result is a stronger and clearer paper. The discussion section in particular is very strong, and the methods are clear. There are a few remaining minor issues.

Although the authors have clarified what they mean by "internal evidence" and "internal representations" (and I do see what they are trying to get at, which is that one can characterize what the network has encoded overall by examining responses on the output layer to various stimuli), I still believe the term 'internal' is potentially confusing in reference to the output layer.

For example, Page 7: "An additional advantage of CNNs is that, unlike humans, we can directly probe the network’s internal representations and understand the mechanisms underlying their behavior." sounds like probing the intermediate/hidden layers and making sense of them, which is practically impossible given the way such networks work, and is not what happened in this work.

Page 9: "As [is] standard in the literature"

Page 12: Do the authors mean "from human studies" instead of "from human behavior"?

Page 21: It would be nice to have some more insight than "interactions between the stimuli and the networks" when the effect isn't obtained. I don't insist, but if the authors have any insight as to why the effect does not always obtain, it might help to include it.

Page 24: I don't understand the statement about "increases in confidence between levels"--perhaps this is a typo?

Page 26: "Underling" should probably be "Underlying"

Page 31: "Further, these manipulations change the properties of the CNNs’ evidence distributions in a way that is consistent with the predictions of an empirically validated model of perception, suggesting that, in the case of the stimuli tested here, a common mechanism can possibly explain the behavior of both humans and ANNs."

The only problem here is that we already know that CNNs fail to predict human behavior in many cases. So the phrase "common mechanism" makes perhaps too strong a commitment to some factor or mechanism internal to the network as opposed to something about the stimulus, a possibility the authors have already pointed out.

Page 32: "However, findings of CNNs failing to replicate human behavior cannot automatically be interpreted as evidence for the necessity of the cognitive mechanism being tested. Differences in behavior between humans and ANNs can also arise from fundamental differences in how their networks process the stimuli

themselves (Wichmann & Geirhos, 2023)." These sentences were very hard to follow.

Page 36: "The possible range of orientations was [0, 2]." switches to radians when everything else was in degrees.

Reviewer #3: The authors have addressed all my concerns. Congratulations to the authors for a very interesting paper.

**Have the authors made all data and (if applicable) computational code underlying the findings in their manuscript fully available?**

Reviewer #1: Yes

Reviewer #2: Yes

Reviewer #3: Yes

PLOS authors have the option to publish the peer review history of their article (what does this mean?). If published, this will include your full peer review and any attached files.

Reviewer #1: No

Reviewer #2: No

Reviewer #3: No

Figure Files:

Data Requirements:

Reproducibility:

References:

---

## [Editor Report · Decision Letter 2]

22 Oct 2024

Dear Dr. Shekhar,

We are pleased to inform you that your manuscript 'Human-like dissociations between confidence and accuracy in convolutional neural networks' has been provisionally accepted for publication in PLOS Computational Biology.

Best regards,

Alex Leonidas Doumas

Academic Editor

PLOS Computational Biology

Andrea E. Martin

Section Editor

PLOS Computational Biology

---

## [Editor Report · Acceptance letter]

5 Nov 2024

PCOMPBIOL-D-24-00636R2 

Human-like dissociations between confidence and accuracy in convolutional neural networks

Dear Dr Shekhar,

I am pleased to inform you that your manuscript has been formally accepted for publication in PLOS Computational Biology. Your manuscript is now with our production department and you will be notified of the publication date in due course.

With kind regards,

Anita Estes
